# Ex Vivo Fluorescence Confocal Microscopy in Specimens of the Liver: A Proof-of-Concept Study

**DOI:** 10.3390/cancers14030590

**Published:** 2022-01-25

**Authors:** Ulf Titze, Karl-Dietrich Sievert, Barbara Titze, Birte Schulz, Heiko Schlieker, Zsolt Madarasz, Christian Weise, Torsten Hansen

**Affiliations:** 1Institute of Pathology, Campus Lippe, University Hospital OWL of the University of Bielefeld, 32756 Detmold, Germany; barbara.titze@klinikum-lippe.de (B.T.); birte.schulz@klinikum-lippe.de (B.S.); torsten.hansen@klinikum-lippe.de (T.H.); 2Department of Urology, Campus Lippe, University Hospital OWL of the University of Bielefeld, 32756 Detmold, Germany; Karl-Dietrich.Sievert@klinikum-lippe.de; 3Department of Gastroenterology, Campus Lippe, University Hospital OWL of the University of Bielefeld, 32756 Detmold, Germany; heiko.schlieker@klinikum-lippe.de; 4Department of General Surgery, Campus Lippe, University Hospital OWL of the University of Bielefeld, 32756 Detmold, Germany; zsolt.madarasz@klinikum-lippe.de; 5Department of Pediatrics, Campus Lippe, University Hospital OWL of the University of Bielefeld, 32756 Detmold, Germany; christian.weise@klinikum-lippe.de

**Keywords:** fluorescence confocal microscopy, liver biopsies, liver tumors, hepatitis, liver transplantation, digital pathology

## Abstract

**Simple Summary:**

Fluorescence confocal microscopy (FCM) is a novel micro-imaging technique providing optical sections of native tissue. The method is increasingly used for the routine diagnosis of tumors and inflammatory lesions of the skin and shows promising results for the diagnosis of other organ tumors. Very few publications exist about examinations of liver tissue thus far. In this study, we compare findings of FCM-examinations of biopsies and surgical specimens of the liver with the final diagnoses from conventional histology.

**Abstract:**

Ex vivo Fluorescence Confocal Microscopy (FCM) is a technique providing high-resolution images of native tissues. The method is increasingly used in surgical settings in areas of dermatology and urology. Only a few publications exist about examinations of tumors and non-neoplastic lesions of the liver. We report on the application of FCM in biopsies, surgical specimens and autopsy material (33 patients, 39 specimens) of the liver and compare the results to conventional histology. Our preliminary examinations indicated a perfect suitability for tumor diagnosis (ĸ = 1.00) and moderate/good suitability for the assessment of inflammation (ĸ = 0.4–0.6) with regard to their severity and localization. Macro-vesicular steatosis was reliably detected, micro-vesicular steatosis tended to be underestimated. Cholestasis and eosinophilic granules in granulocytes were not represented in the scans. The tissue was preserved as native material and maintained its quality for downstream histological, immunohistological and molecular examinations. In summary, FCM is a material sparing method that provides rapid feedback to the clinician about the presence of tumor, the degree of inflammation and structural changes. This can lead to faster therapeutic decisions in the management of liver tumors, treatment of hepatitis or in liver transplant medicine.

## 1. Introduction

Conventional histology is the gold standard for the diagnosis of liver diseases. The classification of tumors as well as the examination of parenchymal diseases, each require high degrees of specialization on the part of the pathologists. Precise knowledge of the entities and an overview of the histological techniques used are necessary for the classification of primary and secondary tumors of the liver. Semi-quantitative grading systems for inflammation, fatty liver disease, allograft rejection and chronic cholestatic diseases were developed based on the systematic assessment of histological features of liver parenchyma. Simple grading and staging systems for chronic hepatitis are appropriate for the management of individual patients, while more complex systems (e.g., Histology Activity Index) are appropriate for the evaluation of large cohorts of patients when statistical analysis is required [1].

Standard pathological assessment is still a time-consuming approach. In recent years, several microscopic methods have been developed for the examinations of unfixed tissue based on modifications of illumination, fluorescence techniques and digital image processing [2,3,4,5]. These new imaging technologies allow for the timely examinations in living patients without tissue removal (in vivo microscopy) or in freshly excised tissue (ex vivo microscopy). Due to their tremendous potential for clinical impact in a wide variety of applications, there has been much effort in recent years to integrate these approaches into pathology practice [6,7,8,9].

Confocal microscopy (CM) is a technique that provides high-resolution images of native tissues for timely examinations in both in vivo and ex vivo approaches [10]. Devices for in vivo examinations, mainly performed in dermatology, ophthalmology and endoscopy [11,12,13], exclusively use reflected laser light (Reflected Confocal Microscopy, RCM). They provide live images of superficial tissue structures in gray scales. Microscopes for ex vivo examinations apply supplemental laser light for fluorescence illumination (Fluorescence Confocal Microscopy, FCM) and require pre-treatment of the tissue with fluorescent dyes. These devices provide digital images that closely resemble hematoxylin-eosin (HE) stained frozen sections.

While some of the applications mentioned above are still experimental, RCM and FCM have been established for the routine diagnostics of neoplastic and inflammatory skin diseases in dermatology [14]. In vivo RCM is used for the early diagnosis of melanocytic lesions [15] and for the follow-up evaluation of therapeutic effects in superficial inflammatory skin diseases [16]. Ex vivo FCM is primarily used for the assessment of surgical margins during Mohs micrographic surgery of skin tumors [17]. Furthermore, recent studies investigate the implementation for the timely intra- and perioperative routine diagnosis of prostate carcinoma in prostate biopsies [18] and the assessment of surgical margins in prostatectomy specimens [19] in urology.

Limited studies exist of FCM examinations for other organ tumors such as the breast, thyroid, colon and lymph nodes [20,21,22]. A recent publication evaluates the feasibility of ex vivo FCM in the analysis of non-neoplastic changes of kidney biopsies [23]. To date, only few publications exist about tumorous lesions and normal tissue and tumors of the liver in general (20 cases in total) [21,24] without defining neoplastic lesions more precisely. The peculiarities of the liver parenchyma and its diseases were not discussed. In particular, the feasibility of ex vivo FCM for the assessment of non-neoplastic parenchymal liver diseases has not yet been examined.

This proof-of-concept study explores the suitability of ex vivo FCM for the examinations of liver specimens. The presented cases cover a representative spectrum of neoplastic and non-neoplastic liver diseases in biopsies, surgical specimens and selected autopsy material. In order to highlight possible applications and limitations of this technology, FCM images were compared to the corresponding conventional histology. We measured the levels of agreement between FCM and conventional histology in the diagnosis of tumors and diagnostic criteria for non-neoplastic liver diseases in a blind setting.

## 2. Materials and Methods

### 2.1. Study Participants

We examined liver biopsies, surgical specimens and autopsy material from 33 patients of the Department of Internal Medicine of the Klinikum Lippe (mean age 64.9 ± 19.5 years, range 3 months to 89 years). The biopsies were obtained either to investigate unclear foci in the liver suspicious for malignancy or for the classification of inflammatory changes in the liver in the context of elevated liver enzymes. We further enrolled specimens from patients who underwent liver surgery in the Department of Abdominal Surgery. Indications for the procedures were masses in the liver and a suspected lesion with Echinococcosis in one case. Furthermore, we examined the liver in an autopsy of a 3 months old infant who succumbed due to septic complications after the implantation of a ventriculo-peritoneal shunt for cerebral bleeding. All study participants (or their relatives) were informed and signed a written consent.

### 2.2. Study Design

We received the biopsies and surgical specimens from the surgical theatre as native material and performed FCM examinations as an intermediate step before subsequent routine histology. The biopsies were thoroughly examined with ex vivo FCM. From the surgical specimen and autopsy material we scanned representative dissections. The FCM scans were blindly evaluated by two experienced pathologists (UT, BT). A third experienced pathologist (TH) blindly evaluated HE sections, special stains, and if necessary, immunohistological stains of the formalin fixed and paraffine-embedded (FFPE) materials as part of the routine diagnostics. A fourth experienced pathologist (BS) blindly re-examined the HE-sections of the FFPE material. Finally, we compared the findings from FCM scans to the results from conventional histology.

### 2.3. Sample Processing and FCM Image Acquisition

The native material was pre-treated with 70% ethanol for 10 s (protein precipitation to enhance contrast) and then incubated with an Acridine Orange solution (AO, 0.6 mM; Sigma-Aldrich^®^, St. Louis, MO, USA) for 30 s [25]. After the staining process has been completed, the tissue samples were manually placed flat on a slide specially prepared with magnets and afterwards covered with a foam pad in order to keep the samples in position and standardize the required distance. A second microscope slide was attached on top of the first slide to hold the tissue in place (Figure 1). The microscope was controlled using a high-power personal computer. Focus-depth and intensity of the illuminating lasers were adjusted in live-view mode. After the adjustments were defined, the specimens were systematically scanned within 2–5 min. The resulting images were then digitally stored as anonymized data for privacy protection.

The specimens were placed in prepared embedding capsules and fixed in 4% PBS-buffered formaldehyde for 24 h immediately after the scanning process. Further histological processing was carried out following the standard procedure for FFPE tissue. Histological diagnoses were established based on HE sections from each of the paraffin-fixed material. Additional special stains (PAS-Diastase, Masson’s trichrome, Elastica van Giesson, Gomori, Iron stain) were routinely obtained for the assessment of glycogen and glycoproteins, collagen and reticulin fibers or iron granules. Immunohistological staining were requested to characterize metastatic tumor. All special stains were performed using the Ventana BenchmarkTM (Ventana Medical Systems, Tucson, AZ, USA) platform. In order to achieve an optimal comparability with the FCM-scan, the biopsies were not removed from the foam pads during subsequent processing.

### 2.4. Ex Vivo Confocal Microscopy

The VivaScope 2500M-G4 (VivaScope, Munich, Germany) used was equipped a water immersion objective with 38× magnification and a numerical aperture of 0.85. Two lasers with wavelengths of 488 nm (blue) and 785 nm (near-infrared) were used for illumination. Both signals were acquired simultaneously and correlated in real-time. AO, that was applied to the tissue prior to the imaging process, was excited by the blue laser (fluorescence mode) highlighting nuclear structures. The near-infrared laser was used to generate the reflectance signal (reflected mode), showing cytoplasmic and extracellular structures. The signals from the fluorescence and reflection channels were saved as gray values in separate images. A built-in algorithm translated both signals into an HE-like pseudo-colored merged image in which the nuclei of the cells were shown in violet, whereas connective tissue fibers and cytoplasm of the cells were shown in pink [26]. The resulting images contained similar information to conventional histology (pseudo-color mode) and could be examined at any desired magnification up to displaying a whole sample at 550-fold magnification. According to manufacturer instructions, samples of up to 2.5 × 2.5 cm in size could be examined.

### 2.5. Data Collection and Statistical Analysis

Both FCM scans (UT, BT) and HE sections (TH, BS) were systematically analyzed by the examiners in a blind test. Each examiner transferred the findings to prepared questionnaires in tabular form. The absence/presence of tumor manifestations were noted using a binary system (0—absent; 1—present). If a tumor was present, its histogenenetic origin was noted (1—epithelial, 2—mesenchymal, 3—other). Non-neoplastic parenchymal changes were systematically assessed based on established metric systems [1]. Nuclear enlargement, ballooning degeneration and signs of apoptosis in hepatocytes (acidophilic degeneration, Councilman bodies, Mallory bodies) were noted in binary systems (0—absent; 1—present) as well as cytoplasmatic, canalicular or perisinusoidal cholate stasis. Inflammatory changes were assessed in terms of the applicability of the Desmet-score [27]. The degrees of portal, periportal and lobular inflammation as well as necrosis were evaluated in four grade rating systems (0—absent, 1—mild, 2—moderate, 3—severe). Fibrotic changes were assessed in regard to their amount (0—normal, 1—mild, 2—moderate, 3—severe) and distribution (0—portal, 1—pericellular, 2—septa, 3—cirrhosis). The absence/presence of granuloma and Kupffer cell/stellate cell hyperplasia was also noted. The biliary system was assessed for cholestatic diseases (absence/presence of bile duct injury, ductular reaction, ductopenia). Steatosis was graded in a semi-quantitative system according to the NAFLD activity score and staging system devised by the Pathology Committee of the NASH Clinical Research Network (0—<5% parenchymal involvement, 1—5–33% parenchymal involvement, 2—33–66% parenchymal involvement, 3—66+% parenchymal involvement) [28].

The histological diagnoses of both pathologists (TH, BS) were statistically compared. Their discordant findings were subsequently assessed and a consensus diagnosis was established. Lastly, the FCM diagnoses from UT und BT were each compared with the consensus HE diagnosis. The individual findings were analyzed in error matrices.

In order to provide comparable results to the previously published results, the levels of inter-observer agreements were measured using the Cohen’s Kappa statistic [29]. The coefficient takes into account the possibility of accidental matches between two raters. The values for measured agreements (p_0_), as well as for the coincidentally expected agreement (p_e_), could be derived from the error matrices without much effort and mathematically related to one another using a formula. Results for κ could vary between κ = 0 for a purely random matches (p_0_ = p_e_) and κ = 1.0 for perfect matches. Negative values for κ implied that there is no effective agreement between the two raters or that the agreement was worse than random. We used the categories according to Landis and Koch for interpretation of Kappa values (κ < 0.00: poor, κ = 0.00–0.20: slight, κ = 0.21–0.40: fair, κ = 0.41–0.60: moderate, κ = 0.61–0.80, κ = 0.81–1.00 almost perfect) [30]. Most previous publications reported very variable levels of interrater-agreement for the individual diagnostic categories [1], which were usually better for fibrosis (kappa 0.5–0.9) than for inflammation (kappa 0.2–0.6) [28,31].

## 3. Results

### 3.1. Histological Diagnoses

The results of our cases enrolled in this study are shown in Table 1. The data set comprised 39 samples (22 biopsies, 16 surgical resections, 1 autopsy specimen) from 33 patients. Both pretherapeutic biopsies and the surgical resections were available from 5 patients. One patient underwent repeated biopsies, because there was no tumor but only tumor necrosis acquired in the first attempt.

Neoplastic lesions were noted in 25 specimens of 19 patients (10 biopsies, 15 resections; 5 patients with biopsy and resection). Malignant primary tumors of the liver were present in 6 patients (4× hepatocellular carcinoma, 2× cholangiocellular carcinoma). Metastases were found in 11 patients (4× neuroendocrine tumors, 3× colorectal cancer, 2× pancreatic cancer, 1× adenocarcinoma of the gallbladder and 1× endometrial carcinoma). Two patients presented with benign primary tumors of the liver (1× hemangioma and 1× focal nodular hyperplasia).

Specimens of 14 patients (12 biopsies, 1 resection, 1 autopsy) contained non-neoplastic inflammatory diseases only. The biopsies of 6 patients presented with lobular inflammatory patterns. Biliary patterns were found in the biopsies of three patients. Another three patients presented with steatotic patterns. In a resection of one patient manifestations of echinococcosis multilocularis with dense perilesional inflammatory infiltrates were demonstrated. Autopsy material of the deceased infant presented findings of neonatal giant cell hepatitis.

Inflammatory changes were also found in the biopsies and surgical specimens of tumor patients. The cases of hepatocellular carcinoma were associated with viral hepatitis showing various degrees and stages of inflammation and fibrosis. Reactive inflammatory lesions were found in the parenchyma adjacent to tumor manifestations.

### 3.2. Assessment of Tumor

Tumor manifestations were reliably detected in the FCM scans by both pathologists. Characteristic histological patterns of primary tumors of the liver were preserved in the FCM scans enabling reliable diagnoses of malignant and benign tumors in the FCM scans. Hepatocellular carcinoma (HCC) could be distinguished from cholangiocellular carcinoma (CCC).

The FCM scans of the cases presenting with HCC demonstrated a good representation of trabecular growth patterns with cell plates three or more cells thick which are very characteristic for these tumor entities (Figure 2A,B). The features of hepatocytic differentiation in the tumor cells were very well preserved. Characteristic clear cytoplasm of the tumor cells in the clear cell variant of HCC were reliably recognizable in FCM (not shown). Mallory bodies, known to be cytoplasmatic inclusions of damaged intermediate filaments, were visible in the tumor cells of one case. These distinct cytoplasmatic changes were barely visible in FCM. Nuclear atypia, representing an important feature for tumor grading, was very well represented in the FCM scans. The degrees of nuclear enlargement and irregularity as well as the presence of prominent nucleoli were reliably assessable.

In cases of intrahepatic CCC (Figure 2C,D), tumor formations were easily visible in FCM scans. Their well to moderately differentiated neoplastic glands were partly easier to recognize in front of the tumor stroma than in conventional histology. The characteristic shape of the tumor cells, that resemble biliary epithelium, was also recognizable in the digital images. Cytological criteria of malignancy, such as marked nuclear atypia with an increased nucleus-to-cytoplasm ratio, increased variation in nuclear size and loss of polarity were well represented in the FCM scans. The inflammatory infiltrates of the desmoplastic stroma were clearly shown.

Benign tumors and masses were recognizable by their localized growth pattern and characteristic histologic features. Focal nodular hyperplasia was correctly diagnosed from the FCM scans. Its typical architecture with incomplete nodules of normal-appearing parenchyma separated by fibrous septa were well represented. The characteristic central scar with characteristic dilated blood vessels could also be identified (Figure 3A,B).

In another case, a hemangioma could be distinguished from a clinically suspected liver metastasis without much effort (Figure 3C,D). Characteristic well-circumscribed large vascular channels, arranged in lobules and lined by flattened endothelial cells were easily identified. The fibrous matrix, which was strongly developed in this case as well as the loosely arranged stromal cells, were clearly visible in the FCM.

Metastases were also reliably detected in the FCM scans (Figure 4). Characteristic growth patterns and glandular morphology of moderately differentiated colorectal and pancreatic adenocarcinoma were shown very well in the digital images. Manifestations of well/moderately differentiated neuroendocrine tumors were identifiable in the scans presenting with preserved trabecular/nested growth patterns and their typical shape of tumor cells. Metastasis of extrahepatic cholangiocellular carcinoma or pancreatic adenocarcinoma showed typical morphologies in FCM. One case of dedifferentiated pancreatic carcinoma presented a sarcomatoid morphology with spindle shaped tumor cells, multinucleated giant cells and high-grade nuclear atypia, that were similarly assessable in the FCM scans. Additional immunohistological examinations were necessary to classify manifestations of metastatic neuroendocrine carcinoma or endometrial carcinoma with less characteristic histologic morphology. Pre-treatment for FCM did not alter immunoreactivity of the tissue (Figure 4, inlays) in all cases.

In summary, all tumors and masses of the liver were correctly recognized and classified in the digital scans. Ratings from FCM-examination showed almost perfect levels of agreement with the diagnosis in conventional histology.

### 3.3. Assessment of Parenchymal Changes

The degree of inflammatory infiltrates could be reliably determined in FCM and assigned to the relevant compartments of the tissue (Figure 5). For the representation of the cell nuclei, particularly the short-wave excitation laser needed to be adjusted precisely in order to prevent too strong or too weak signals. Given proper adjustments, granulocytes, plasma cells and lymphocytes could be distinguished. Remarkably, the characteristic granules of eosinophil granulocytes were not observed in the reflected or fluorescence mode (Figure 5D,E). Stellate cells were reliably recognizable. In cases of NASH, inflammatory foci were represented in FCM. Granuloma, nodules of Kupffer cells and endothelialitis could not be observed in the present cases. Over all, we found moderate to substantial levels of agreement between the FCM-scans and HE-diagnoses for the evaluation of inflammatory infiltrates in a four-tired grading system.

As in conventional HE-stains, lower degrees of fibrosis were not visible and could be only detected in specific trichrome stains in FFPE processed material (Figure 5A–C). Advanced degrees of fibrosis with septa and cirrhosis could be classified analogously to conventional HE morphology and fibrous stroma reactions in benign and malignant tumors could be recognized.

The lipid content of the parenchyma could be estimated in FCM with good agreement to histology. Macrovesicular steatosis was very well represented in the digital scans. In contrast, microvesicular steatosis was difficult to detect in the FCM. Over all, the degree of steatosis tended to be slightly underestimated based on FCM (average rating 0.51 in conventional histology vs. 0.45 in FCM). Nevertheless, the cases with clinically relevant steatosis were reliably identified.

Ductular changes in cases with biliary patterns were visual in FCM. Ductular proliferations were recognizable with moderate levels of inter-observer reliability between FCM and HE diagnoses. In contrast, the degree of acute bile duct injury was under-represented in the FCM-ratings.

Cytoplasmatic changes in hepatocytes appeared to be elicitable only to a limited extent in FCM (Figure 6). In normal tissue, the hepatocytes could be identified without any doubt on the basis of their shape. Reactive changes such as nuclear enlargements, presence of multinucleated hepatocytes and nuclear inclusions were well represented. Pathologic changes of cytoplasm were very subtle in the FCM scans and could only be identified retrospectively in comparison with the corresponding HE slide. Mallory bodies, known to be cytoplasmatic aggregates of damaged intermediary filaments in hepatocytes, were represented only as very discrete signal-rich areas of cytoplasm (Figure 6C,D). Ballooning degeneration hepatocytes is another characteristic finding in alcoholic steatohepatitis, presenting as enlargement of rounded hepatocytes with cobweb-like cytoplasm. The affected hepatocytes are conspicuous in FCM because of their size, whereas the cytoplasmic changes are hardly formed. Cytoplasmatic or canalicular bile was not represented in reflected or fluorescence signals appearing as optically empty vacuoles in the FCM scans (Figure 6A,B).

### 3.4. Additional Findings

The lesions of Echinococcus multilocularis presented as a characteristic fibrotic mass with necrosis, severe perilesional inflammation and multiple intra- and perihepatic daughter cysts of variable sizes (Figure 7A,B). While the inner layers and protoscolices were not present after therapy, the pathognomonic middle layers and outer inflammatory infiltrates were easily recognizable in FCM. Characteristic laminated acellular material of the middle layer secreted by the parasite was represented both in the reflected and fluorescence mode (Figure 7A inlay).

### 3.5. Reproducibility of Diagnostic Features

The orienting statistical analysis showed that liver tumors were detected in ex vivo FCM with high levels of agreement to conventional histology (kappa values 1.0). Histogenetic classification of tumors was also in high agreement with FFPE morphology (kappa values 0.83). Only one neuroendocrine tumor was misinterpreted as a mesenchymal or lymphoid tumor in FCM.

In the literature, the histological criteria of non-neoplastic changes show very different grades of reproducibility even in conventional histology (kappa values 0.4–0.6 for portal inflammation, interface hepatitis and parenchymatous inflammation, kappa values > 0.6 fibrosis grade, steatosis, cholestasis as well as cytoplasmic changes such as ballooning degeneration and Mallory bodies). In this study, analogous kappa values were found for the FFPE ratings (TH vs. BS, Table 2).

Moderate to substantial levels of agreement between FCM and conventional histology were found for portal inflammation (kappa 0.6/0.6), lobular inflammation (kappa 0.5/0.7), and interface hepatitis (kappa 0.5/0.4). Moderate agreement was also found for steatosis (kappa 0.6/0.6) and degree as well as distribution of fibrosis (kappa 0.6–0.7). Simile degrees of reproducibility were found for ductular proliferations (kappa 0.5/0.5) and stellate cell hyperplasia (kappa 0.6/0.6). The lack of or very discrete representation of cytoplasmic and biliary changes were also reflected in the ratings. For the detection of Mallory bodies and bile stasis, the kappa values strongly differed (0–0.1 in FCM vs. 0.6/0.5 in conventional histology).

## 4. Discussion

In our opinion, FCM is a very promising tool for fast evaluations of biopsies and surgical specimens of the liver. This study provided a basis for future researchers in the field of abdominal surgery, gastroenterology and especially for transplantation to explore and advance the potential application of this technology. Preserved immunoreactivity of the tissue indicates that previous treatment for FCM does not cause any limitations for subsequent immuno-histological or molecular examinations. This is explained by the low penetration depth of AO at an incubation time of 30 s which we found in our previous examinations of biopsy material. We previously demonstrated that DNA content was not significantly diminished after pre-treatment for FCM [32]. Therefore, any necessary immuno-histological or even molecular examinations of diagnostically difficult tumors do not constitute an obstacle to preliminary examinations in FCM.

Although the impact of liver biopsies underwent major changes in the last decade, the emergence of new technologies for histologic evaluation, tissue content analysis and genomics hold great promise for the future and might shape the indications for biopsy acquisition [33]. Ex vivo FCM can represent another part of this mosaic of digital pathology as specimen preparation is simple and digital images are provided that enable online remote interpretations by trained pathologists and/or specialized clinicians. Larger series of digital FCM images might be a suitable basis for building neural networks in the future that contain data from conventional histology, immunohistology, molecular analyzes, MRI findings and clinical presentations as well as clinical outcome.

According to previous evaluations in other organs, tumor manifestations of the liver were reliably diagnosed. False-negative findings are to be expected in up to 5–10% of ultrasound-guided tumor biopsies [34]. Using FCM, timely feedback can be provided to the clinician as to whether a malignant tumor is present and representative tumor material has been obtained for subsequent immune-histological and even molecular examinations. Rapid diagnosis allows any necessary repeat biopsies or supplemental imaging to be initiated during the current hospital stay. The role of liver biopsies in the management of hepatocellular carcinoma is controversial; however, the option to obtain tumor material and preserve it as fresh tissue for future molecular analysis in personalized medicine is an important prospect [35]. Our findings also indicated a good feasibility of this technique for intraoperative settings. In analogy to its application in the surgical treatment of prostate cancer [18], FCM might be an interesting alternative to frozen sections for the intraoperative assessment of surgical margins in the future.

Our evaluations of non-neoplastic parenchymal changes showed that the grade and the distribution of inflammatory infiltrates were assessed with acceptable levels of agreement to conventional histology. Furthermore, advanced degrees of fibrosis and the amount of steatosis were reliably detected. In analogy with experience from dermatology [16], this indicated a good suitability of FCM for assessing therapeutic effects on chronic hepatitis in follow-up biopsies. The performance in this setting should be evaluated in systematic studies for the individual entities in larger biopsy series. However, limitations in the assessment of biliary stasis and in the representation of cytoplasmatic features in hepatocytes and inflammatory infiltrates indicated that the method will not be able to completely replace conventional histological processing. The etiologic classification of chronic hepatitis should still be reserved to examinations of FFPE -processed material with special stains.

Ex vivo FCM is also a very interesting tool for timely examinations of biopsies from liver transplants. Its feasibility to provide timely feedback on the extent of macro-vesicular steatosis, which is an accepted adverse prognostic factor after liver transplantation [36], makes the system a promising alternative to frozen sections of pretransplant donor biopsies [37,38]. Digital FCM scans of the biopsies could be provided within 20–25 min for online evaluation to specialized pathologists worldwide leading to faster decisions on the suitability of the explanted organs. Since the degree of steatosis tended to be underestimated in our evaluations on native material, further tests are indicated with varying degrees of tissue pre-treatment with alcohol or alternative solvents.

FCM also appears to be a promising tool for the diagnosis of transplant rejection. The Banff criteria for acute and chronic rejection primarily include the extent of inflammatory infiltrates and change in blood vessels and bile ducts [39,40]. It would be interesting to study in larger series to what extent the criteria for chronic and acute rejection can be assessed in the FCM scans. The digital images could provide a suitable basis for neural networks that would allow standardized analysis of graft biopsies.

## 5. Conclusions

Our preliminary work demonstrated that FCM allows histological examinations of unfixed liver samples within half an hour after biopsy acquisition. The diagnostic power is similar to frozen sections and allows reliable tumor diagnoses and statements on the extent of inflammatory infiltrates or structural changes, especially macro-vesicular steatosis. In contrast to conventional histology, sample preparation is simple and can be performed manually within a few minutes. The digital scans are ready to be assessed by local pathologists after 10 min. This enables more effective diagnosis of tumors, as any subsequent additional examinations can be initiated in a timely manner. The images are available as digital data and, with the appropriate infrastructure, can be provided for online examination by specialists worldwide. This may open up new perspectives in transplant pathology, especially for the diagnosis of pre-transplant donor biopsies.

## Figures and Tables

**Figure 1 cancers-14-00590-f001:**
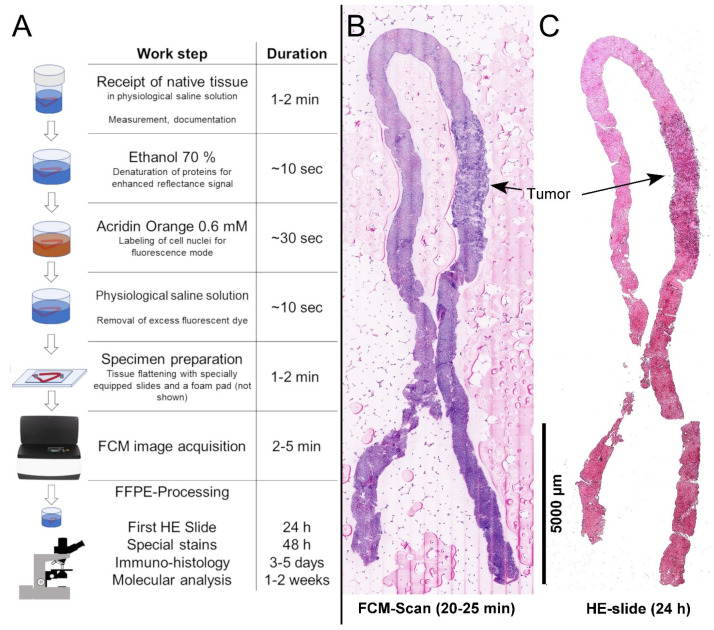
Processing of the native material for FCM. (**A**): The native material was transferred into ethanol, AO and saline solution and finally positioned on a slide specially equipped with magnets. The tissue was spread evenly on the microscope slide using an additional foam pad (not shown). This process to prepare the native material for confocal microscopy was simple and took approximately 3–4 min. Depending on the size of the material examined, the scan took between 2–5 min. The resulting digital image was viewed approximately 10 min later. (**B**,**C**): Matching visualization of the tissue in FCM (**B**) and conventional histology (**C**) is seen. The tumor infiltrate detected in the biopsy is easily recognizable in both images.

**Figure 2 cancers-14-00590-f002:**
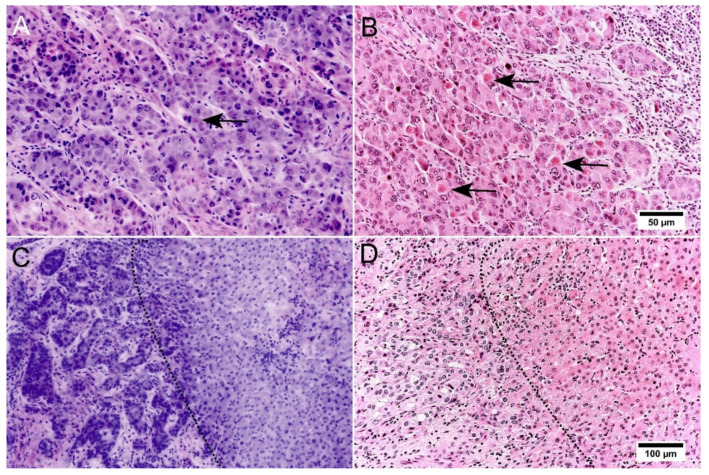
Representation of malignant primary tumors of the liver in FCM scans. (**A**,**B**): Hepatocellular carcinoma (grade 2) in FCM (**A**) and conventional histology ((**B**), HE staining). Infiltrative growth and trabecular architecture were easily recognizable in FCM scans. Note the missing representation of Mallory bodies (arrows) in the FCM scans in comparison to the HE-slide. (**C**,**D**): Intrahepatic cholangiocellular carcinoma of the liver in FCM (**C**) and conventional histology ((**D**), HE staining). Infiltrating groups of neoplastic glands were reliably recognizable in FCM. The dashed line marks the invasion front between tumor formations (**left**) and pre-existing liver parenchyma (**right**).

**Figure 3 cancers-14-00590-f003:**
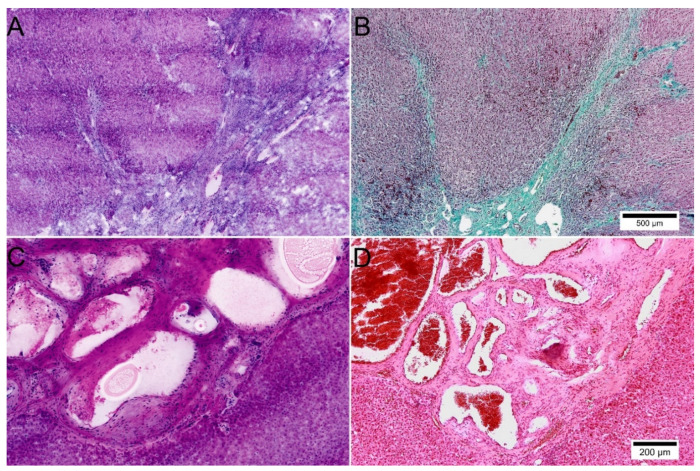
Representation of benign primary tumors and masses of the liver in FCM scans. (**A**,**B**): Focal nodular hyperplasia in FCM (**A**) and conventional histology with special staining ((**B**), Masson’s trichome). Note the radial scar and fibrous septa are well represented in FCM. (**C**,**D**): A case of hemangioma in FCM (**C**) and conventional histology ((**D**), HE staining). Good representation of vessels and a fibrous matrix in the FCM scans. Note the lining of unsuspicious endothelial cells in FCM.

**Figure 4 cancers-14-00590-f004:**
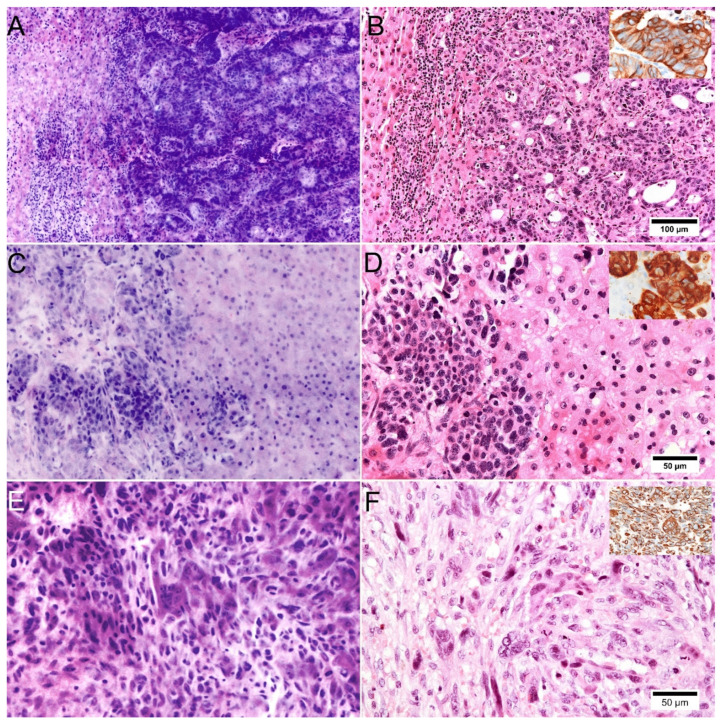
Representation of metastatic neoplasms of the liver in FCM scans. (**A**,**B**): Metastasis of colorectal carcinoma in FCM (**A**) and conventional histology ((**B**), HE staining). Typical glandular patterns of the primary tumor were preserved in this case and clearly recognizable in the FCM scans. Characteristic expression of Cytokeratin 20 (inlay) was preserved in the material. (**C**,**D**): Liver biopsy with focal manifestations of poorly differentiated neuroendocrine carcinoma in FCM (**C**) and conventional histology, ((**D**), HE staining) showing no characteristic features. Immunohistology showed preserved expression of neuroendocrine markers (Chromogranin A, inlay). (**E**,**F**): Metastasis of dedifferentiated pancreatic carcinoma in FCM (**E**) and conventional histology ((**F**), HE staining) presenting sarcomatoid morphology. Immunohistology (Vimentin, inlay) was not helpful in this case. The final diagnosis was established with knowledge of the pancreatoduodenectomy specimen.

**Figure 5 cancers-14-00590-f005:**
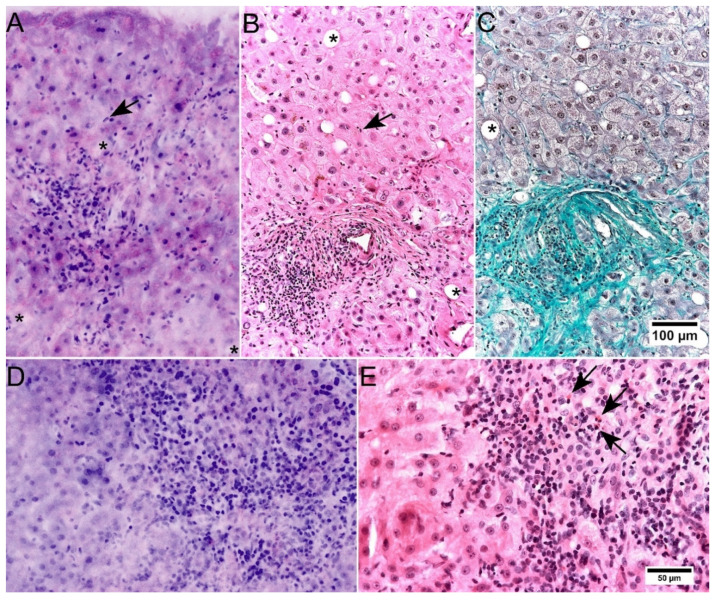
Representation of inflammatory changes in FCM scans. (**A**–**C**): Liver biopsy of a case with methotrexate induced liver injury in FCM (**A**) and conventional histology ((**B**), HE staining) with special stains ((**C**), Masson’s trichrome). Portal and interface inflammation are well represented in FCM. Stellate cell hyperplasia (arrows) can be recognized in FCM comparable to conventional histology. Fatty vacuoles (asterisks) are more difficult to recognize in FCM. Fibrotic fibers are not represented in FCM and HE stain, the real amount of fibrosis is only recognized in Masson’s trichrome staining. (**D**,**E**): Liver biopsy of a case with drug induced liver injury in FCM (**D**) and conventional histology ((**E**), HE staining). Portal and periportal inflammatory infiltrates are shown in a comparable way. Note the missing representation of eosinophilic granules in the FCM scans (arrows).

**Figure 6 cancers-14-00590-f006:**
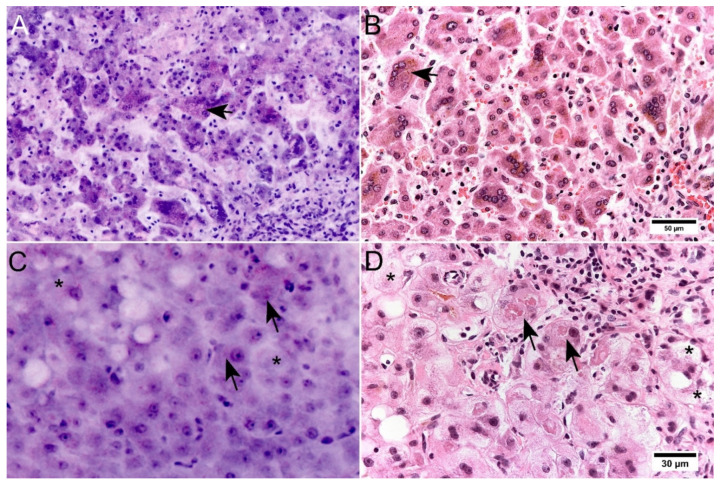
Limited representation of cytoplasmatic changes in FCM scans. (**A**,**B**): Neonatal giant cell hepatitis in FCM (**A**) and conventional histology ((**B**), HE staining). FCM scans show a marked lobular inflammatory infiltrate and trabeculae of multi-nucleated hepatocytes. Note the discrepancy in the presentation of cytoplasmic bile accumulation. In FCM, cytoplasmatic accumulation of bile (arrows) does not give a signal and is only represented as empty cytoplasmic vacuoles. (**C**,**D**): A Case with alcoholic steatohepatitis in FCM (**C**) and conventional histology ((**D**), HE staining). Macrovesicular steatosis is very well represented in FCM scans of native liver tissue. Ballooning degeneration (asterisks) and Mallory bodies (arrows), that are clearly visible in the HE-slides, are very subtle in FCM scans and were missed by the blind assessment. There is also no correlate for canalicular bile stasis visible in the HE-slide.

**Figure 7 cancers-14-00590-f007:**
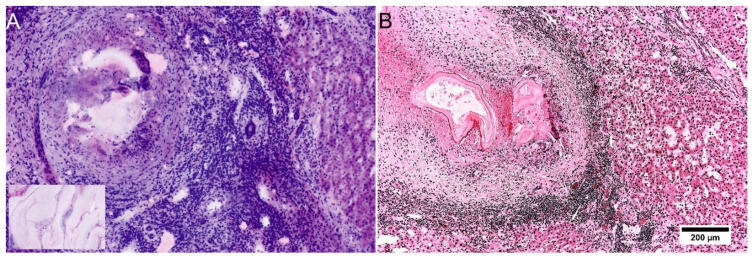
Manifestations of Echinococcus multilocularis in FCM scans and conventional histology. A/B: Echinococcus multilocularis in FCM (**A**) and conventional histology ((**B**), HE staining). One of multiple daughter cysts are shown in the pictures with characteristic layers. The outer layer contains granulation tissue with marked inflammatory infiltrates and foreign body reaction. Characteristic laminated acellular material of the middle layer is represented also in FCM scans (inlay).

**Table 1 cancers-14-00590-t001:** Clinical data and diagnoses from histological evaluation of the specimens.

Patient	Sex	Age	Specimens	Histological Diagnosis	
P01	female	82 years	B	Metastasis	Endometrial carcinoma
P02	male	75 years	B	Inflammatory	Toxic injury (methotrexate)
P03	female	77 years	B + R	Primary Tumor	Cholangiocellular carcinoma
P04	male	3 months	A	Inflammatory	Neonatal syncytial giant cell hepatitis
P05	male	89 years	B (2×)	Metastasis	Tumor necrosis; neuroendocrine tumor (grade 2)
P06	female	83 years	B	Metastasis	Neuroendocrine tumor (grade 1)
P07	male	86 years	R	Metastasis	Colorectal carcinoma
P08	male	80 years	R	Primary Tumor	Hepatocellular carcinoma (grade 2)
P09	male	44 years	B	Inflammatory	Toxic injury (polytoxicomanic patient)
P10	female	52 years	R	Inflammatory	Echinococcus multilocularis
P11	female	79 years	B	Metastasis	Neuroendocrine tumor (grade 3)
P12	male	68 years	B + R	Primary Tumor	Cholangiocellular carcinoma
P13	male	42 years	B	Inflammatory	IgG4-associated cholangitis
P14	male	78 years	R	Metastasis	Neuroendocrine tumor (grade 2)
P15	male	64 years	B	Inflammatory	Toxic injury (drugs)
P16	male	78 years	R	Metastasis	Adenocarcinoma of the gallbladder
P17	female	59 years	R	Primary Tumor	Hemangioma
P18	female	36 years	B	Inflammatory	Ascending cholangitis
P19	male	52 years	B	Inflammatory	Toxic injury (drugs)
P20	female	81 years	B + R	Primary Tumor	Hepatocellular carcinoma (clear cell type, grade 2)
P21	female	57 years	R	Metastasis	Pancreatic carcinoma (Adenocarcinoma)
P22	male	68 years	B + R	Primary Tumor	Hepatocellular carcinoma (grade 1)
P23	male	70 years	B	Inflammatory	Steatohepatitis
P24	male	60 years	R	Metastasis	Colorectal carcinoma
P25	male	74 years	B + R	Primary Tumor	Hepatocellular carcinoma (grade 3)
P26	female	81 years	B	Inflammatory	Toxic injury (drugs)
P27	male	83 years	R	Metastasis	Pancreatic carcinoma (sarcomatoid differentiation)
P28	female	59 years	B	Inflammatory	Alcoholic steatohepatitis
P29	male	33 years	B	Inflammatory	Non-alcoholic steatohepatitis (NASH)
P30	male	70 years	B	Inflammatory	Toxic injury (azathioprine)
P31	female	51 years	R	Primary Tumor	Focal nodular hyperplasia
P32	male	20 years	B	Inflammatory	Primary sclerosing cholangitis
P33	male	80 years	R	Metastasis	Colorectal carcinoma
**Patients’ Characteristics**					** *N* ** **= 33**
Age					64.8 ± 19.4 years (range: 3 months–89 years)
male: female					21/12
**Specimens**					** *N* ** **= 39**
Biopsies (B)					22
Surgical resections (R)					16
Autopsy (A)					1

**Table 2 cancers-14-00590-t002:** Reproducibility of histopathologic features.

Level of Agreement	Diagnostic Criterion	Kappa (UT)	Kappa (BT)	Kappa (TH/BS)
Almost perfect to perfect	Tumor	1.0	1.0	1.0
Kappa > 0.8	Histogenesis of tumor	0.8	0.8	1.0
	Nuclear enlargement	0.7	0.5	0.5
	Steatosis	0.6	0.6	0.8
	Portal inflammation	0.6	0.6	0.6
Moderate to substantial	Periportal inflammation (interface hepatitis)	0.5	0.4	0.6
Kappa = 0.4–0.8	Lobular inflammation	0.5	0.7	0.5
	Stellate cell hyperplasia	0.6	0.6	0.6
	Ductular reaction	0.5	0.5	0.4
	Quantity of fibrous tissue	0.6	0.7	0.7
	Location of fibrosis	0.6	0.5	0.6
	Mallory bodies	−0.1	0.0	0.6
	Bile stasis	0.1	0.1	0.5
	Ballooning degeneration	0.1	0.1	0.2
Slight to fair	Acidophilic degeneration/apoptosis	0.1	0.3	0.2
<0.4	Dropout/confluent necrosis	0.0	−0.1	0.0
	Bile duct injury	−0.1	0.1	−0.2

## Data Availability

The data presented in this study are available on reasonable request from the corresponding author. The data are not publicly available due to privacy restrictions.

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
