# Peer review of "Ex Vivo Fluorescence Confocal Microscopy in Specimens of the Liver: A Proof-of-Concept Study"

_cancers, 2022, doi:10.3390/cancers14030590_

Round 1

Reviewer 1 Report

This paper represents a proof of concept of ex vivo fluorescence confocal microscopy in surgical specimens and diagnostic biopsies of the liver. The findings are really helpful in this field. I found something to improve before publication.

Please show us comparison between FCM and conventional HE using the same or adjacent section(s). It is really good to understand the benefit of this method.

Please check English again.

Table 1 needs to be improved.

Author Response

This paper represents a proof of concept of ex vivo fluorescence confocal microscopy in surgical specimens and diagnostic biopsies of the liver. The findings are really helpful in this field.
I found something to improve before publication.
1. Please show us comparison between FCM and conventional HE using the same or adjacent section(s). It is really good to understand the benefit of this method.
****
Answer: Another figure (Figure 1) added showing the individual steps and their duration. A complete FCM scan can be compared with the corresponding histological slide. It takes about 3-4 minutes to prepare the native material for FCM. Depending on the size of the examined material, the scan takes between 2-5 minutes. The resulting digital image can be viewed after about 10 minutes. In all cases, the preliminary result was provided to the clinician not later than 20-25 minutes.
****
2. Please check English again.
****
Answer: English revisions performed by a native speaker editor (see acknowledgments).
****
3. Table 1 needs to be improved.
****
Answer: Expanded Table 1. The patients are now listed individually. It is now easier to see that in 5 patients both biopsy and surgical specimens were available. One patient underwent repeated biopsy because there was only tumor necrosis in the first biopsy.
****

Reviewer 2 Report

The authors present an initial study on the feasibility of confocal microscopy for ex vivo analysis of resected liver tissues, an organ which has been rather neglected in the literature to date. The manuscript is well organized and the paper is very well written. Regarding the scientific content of the manuscript, I have no issues and no major suggestions. I think the paper is acceptable for publication as-is. My only question is whether "Cancers" is the right outlet for this work, or if it would be better submitted to a journal more targeted to a pathologist readership. This is a question for the authors and editors to consider.

Author Response

Thank you very much for your benevolent comment!

Reviewer 3 Report

Please see the attached .pdf file.

Round 2

Reviewer 1 Report

Now, this paper is fine.

Author Response

Thank you for your helpful collaboration on this article.

Reviewer 3 Report

The authors have improved the manuscript significantly, highlighting and quantifying the benefits of complementing conventional (histopathological) analyses and diagnoses with the introduced FCM method. The methodology is now clearly represented throughout the manuscript with appropriate estimates of its advantages (e.g. time savings). The new schematic as Figure 1 is also a clear improvement and serves as a quick roadmap to the FCM workflow. Taken together, all my other concerns have also been meticulously addressed in the rebuttal letter as well as in the revised manuscript. The text reads well, the figures have undergone improvements, and the overall clarity is better. I have two minor comments that I noticed upon reading through the article:

Line 305: In Figure 4 caption there are two "in" prepositions, please correct.

Line 313: I believe that the in the section "3.3. Assessment of parenchymal changes" there is a slight confusion with the figure numbering. I.e. starting line 313 there should be description of Figure 5 (a few occasions), and starting line 347 (Cytoplasmatic changes in hepatocytes ...) Figure 6 should be discussed instead of Figure 5. If this is the case, please revise and check again that the main text annotations and figure numbering are consistent.

After addressing these minor points, I am glad to advocate the publication of this work in a top-tier journal such as Cancers represents.

Author Response

Again, thank you for your conscientious review of the manuscript. We appreciate your thorough work and constructive comments, which undoubtedly helped to improve the article. Many thanks and best regards